# Stochastic Order Learning: An Approach to Rank Estimation Using Noisy Data

## Abstract

A novel algorithm, called stochastic order learning (SOL), for reliable rank estimation in the presence of label noise is proposed in this paper. For noise-robust rank estimation, we first represent label errors as random variables. We then formulate a desideratum that encourages reducing the dissimilarity of an instance from its stochastically related centroids. Based on this desideratum, we develop two loss functions: discriminative loss and stochastic order loss. Employing these two losses, we train a network to construct an embedding space in which instances are arranged according to their ranks. Also, after teaching the network, we identify outliers, which are likely to have extreme label errors, and relabel them for data refinement. Extensive experiments on various benchmark datasets demonstrate that the proposed SOL algorithm yields decent rank estimation results even when labels are corrupted by noise.

## 1 Introduction

Rank estimation aims to predict the rank or 'ordered class' of an object, which is a fundamental problem in machine learning. It has a variety of applications, such as facial age estimation (Ricanek & Tesafaye, 2006; Shin et al., 2022), aesthetic score regression (Kong et al., 2016), and medical assessment. Many approaches have been proposed for rank estimation (Pan et al., 2018; Li et al., 2021; Liu et al., 2018). Recently, order learning techniques (Lim et al., 2020; Shin et al., 2022; Lee et al., 2022) have shown promising results.

Although these techniques provide remarkable results, their performance may be sub-optimal under the presence of label noise, as they are trained based on the assumption of accurate label annotations. In many real-world scenarios, however, it is quite challenging to obtain error-free annotations of 'ordered data.' Unlike nominal data, it is difficult to distinguish between classes of ordered data, since there is often no clear distinction between ranks. Figure 1(a) and (b) show nominal data for classification and ordered data for rank estimation, respectively. Note that predicting the exact rank of an instance is hard even for humans.

Many algorithms have been developed to train machines using imperfect data with noisy labels, but most of them are for classification (Yao et al., 2022; Ye et al., 2023) or segmentation (Yang et al., 2020). Unlike classification, different errors have different severities in rank estimation. As classification-based approaches do not consider this ordinal property, they yield poor performances when applied to noisy rank estimation. Therefore, it is crucial to develop a noise-robust algorithm specialized for the task of rank estimation.

In this paper, we propose a novel algorithm, referred to as stochastic order learning (SOL), to estimate ranks reliably under the existence of label noise. Given a training dataset with noisy labels, we first model the label errors with random variables. Hence, each instance relates stochastically to multiple ranks rather than deterministically to a single rank. We then train an embedding network based on a desideratum, which encourages minimizing stochastic dissimilarities of instances from their corresponding centroids. To achieve this, we develop the discriminative loss and the stochastic order loss. Moreover, after the training, we identify outliers that are likely to have extreme label errors and relabel them to refine the noisy dataset. Extensive experiments demonstrate that the proposed SOL algorithm provides reliable rank estimation results on various ordered datasets. Furthermore, SOL even reduces the overall label noise of a given dataset based on the outlier detection and relabeling.

(a) Label noise in classification  (b) Label noise in rank estimation

Figure 1: Nominal data in classification versus ordered data in rank estimation. True labels are inside parentheses.

The contributions of this paper can be summarized as follows.

- We extend the concept of order learning to cope with noisy data by designing a stochastic approach; we model label errors as random variables and derive embedding space constraints to sort instances according to their stochastically related ranks.
- Also, we propose outlier detection and relabeling schemes to identify instances with extreme label errors and reduce the overall noise level of a given dataset.
- Experiments on various benchmark datasets for facial age estimation, aesthetic score regression, and medical image assessment validate the effectiveness of the proposed SOL under label noise.

## 2  RELATED WORK

**Learning from noisy labels:** With the availability of substantial training data, deep learning has shown impressive performance in numerous machine learning tasks, but the performance may degrade severely in the presence of label noise. Thus, learning from noisy labels has been an active area of research; various attempts have been made to alleviate the adverse impacts of label noise. Some are based on robust loss functions (Ghosh et al., 2017; Zhang & Sabuncu, 2018; Lyu & Tsang, 2019; Ma et al., 2020; Ye et al., 2023). Other methods include regularization (Tanno et al., 2019; Menon et al., 2020; Xia et al., 2020), robust network architecture (Han et al., 2018a; Goldberger & Ben-Reuven, 2022), and selective data sampling (Han et al., 2018b; Jiang et al., 2018; Song et al., 2019). These methods, however, focus on classification or segmentation tasks.

Compared to classification, only a few noise-robust regression algorithms have been developed (Garg & Manwani, 2020; Liu et al., 2020; Yao et al., 2022; Liu et al., 2024). Garg & Manwani (2020) first considered label noise in ordinal regression. They, inspired by Natarajan et al. (2013), proposed an unbiased estimator approach, in which loss correction takes place to cope with class-dependent label noise. Liu et al. (2020) designed a label regularization strategy to suppress possible noise in ordinal datasets. Yao et al. (2022) developed a variant of Mixup (Zhang et al., 2018), which improves generalization performance by training on virtual examples linearly interpolated from two training samples. To make the Mixup technique more suitable for regression tasks, they sampled a pair with closer ordinal labels with a higher probability. Recently, Liu et al. (2024) proposed a robust algorithm for support vector ordinal regression. Even though their algorithm addresses label noise explicitly, it only handles cases in which the number of ordinal labels is less than 5. In contrast, the proposed algorithm can deal with more difficult settings where datasets have more than 50 ranks.

**Rank estimation:** Different from ordinary classification, rank estimation aims to predict the ordered class of an object. Many rank estimation methods estimate object ranks directly using classifiers or regressors. Early methods convert a rank estimation problem into multiple binary classification problems (Frank & Hall, 2001; Li & Lin, 2006). Recently, several techniques have been developed to perform deep ordinal regression more effectively, such as pairwise regularization (Liu et al., 2018), soft labels (Diaz & Marathe, 2019), continuity-aware probabilistic network (Li et al., 2019), and uncertainty-aware regression (Li et al., 2021).

**Order learning:** Order learning (Lim et al., 2020) is a new approach to rank estimation based on the idea that relative assessment is easier than absolute assessment. Instead of direct prediction, Lim et al. (2020) estimated the rank of an instance by comparing it with references of known ranks.

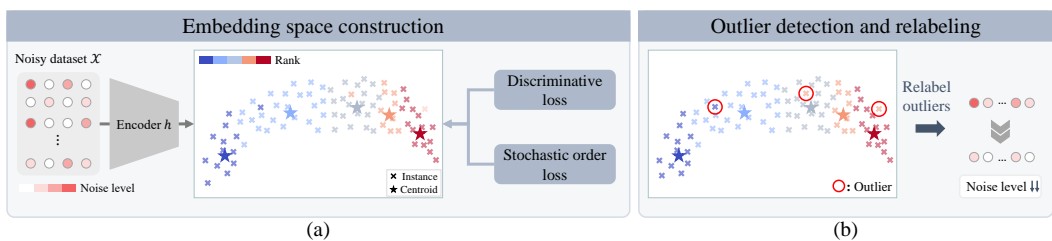

Figure 2: Overview of the proposed SOL algorithm

To find more reliable references, Lee & Kim (2021) proposed the order-identity decomposition network. Shin et al. (2022) extended the idea of order learning to regression problems, and Lee & Kim (2022) and Lee et al. (2024) developed weakly-supervised and unsupervised techniques for order learning, respectively. Also, Lee et al. (2022) proposed a learning mechanism that exploits not only ordering relations but also metric information among object instances. Similar to the proposed algorithm, they constructed an embedding space in which objects are sorted according to their ranks. However, their algorithm assumes that rank labels are deterministic and error-free, so it fails to model the uncertainty and noise in data. To construct a well-arranged embedding space even in the case of label noise, we propose a stochastic approach called SOL in this paper.

## 3 PROPOSED ALGORITHM

### 3.1 PROBLEM DEFINITION

There is a training set $\mathcal{X}$, whose each instance is attributed with one of the $n$ ranks (or ordered classes), represented by consecutive integers in $\{1, \ldots, n\}$. Let $\bar{r}_x$ denote the true rank of instance $x \in \mathcal{X}$. However, only a noisy rank $r_x$ is available, given by

$$r_x = \bar{r}_x + e_x \tag{1}$$

where $e_x$ is the label error of $x$. Let $\mathbf{e}$ be the random variable underlying each error $e_x$. It is assumed that $\mathbf{e}$ has a discrete Gaussian distribution;

$$p_s \triangleq \Pr(\mathbf{e} = s) = \frac{1}{C} e^{-\frac{s^2}{2\sigma^2}} \tag{2}$$

where $C = \sum_t e^{-\frac{t^2}{2\sigma^2}}$, and $s, t \in \mathbb{Z}$. Note that the noise distribution in (2) is symmetric ($p_s = p_{-s}$) and unimodal ($p_s \geq p_t$ for $0 \leq s \leq t$). This models label errors in practice. For example, it is more likely for an annotator to mislabel a 10-year-old as 8 or 12 years old than as 20 years old.

Given the noisy training set $\mathcal{X}$, the objective is to develop a neural network to estimate the ranks of unseen test instances reliably. To this end, we propose SOL. Furthermore, we propose detection and relabeling schemes for outliers, *i.e.* instances with extreme label errors $e_x$.

### 3.2 STOCHASTIC ORDER LEARNING

We employ an encoder $h$ to map each instance $x \in \mathcal{X}$ into a feature vector $h_x = h(x)$ in an embedding space, as shown in Figure 2. We aim to construct the embedding space in which the instances are arranged according to their ranks and each 'centroid' $\mu_r$ is the representative vector for instances with rank $r \in \{1, \ldots, n\}$. However, since only the noisy rank $r_x$ in (1) — instead of the true rank $\bar{r}_x$ — is available, instance $x$ relates stochastically to multiple centroids, rather than deterministically to the single centroid $\mu_{\bar{r}_x}$. More specifically, $x$ is associated with $\mu_{r_x+s}$ with probability $p_s$ in (2). Thus, in the embedding space, the weighted sum of squared distances $\sum_s p_s d^2(h_x, \mu_{r_x+s})$ should be minimized, where $d$ denotes the Euclidean distance.

Hence, we define the stochastic dissimilarity of instance $x$ from rank $r$ as

$$D(x, r) = \sum_s p_s d^2(h_x, \mu_{r+s}). \tag{3}$$

During the encoder training, given each instance $x$ with the noisy label $r_x$, we have the following desideratum:

$$D(x, r_x) \leq D(x, r) \quad \text{for all } r \in \{1, \ldots, n\}. \tag{4}$$

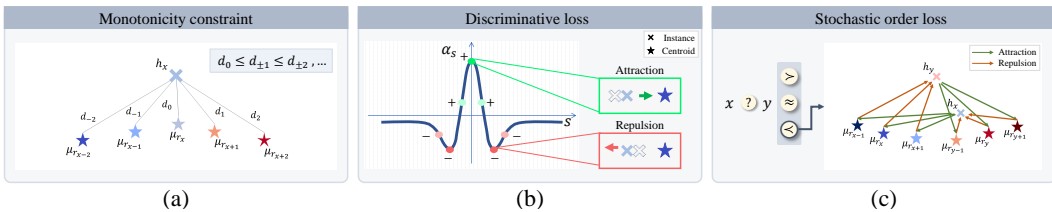

Figure 3: Illustration of constraint and training losses for construction of embedding space

Also, we determine each centroid $\mu_r$ to minimize $\sum_{x \in \mathcal{X}} D(x, r_x)$ based on the desideratum,

$$\mu_r = \frac{\sum_{x \in \mathcal{X}} p_{r-r_x} h_x}{\sum_{x \in \mathcal{X}} p_{r-r_x}}, \quad r \in \{1, \ldots, n\}, \tag{5}$$

as derived in Appendix A. We update the centroids after every training epoch.

A sufficient condition for satisfying the desideratum in (4) is the monotonicity constraint, given by

$$d(h_x, \mu_{r_x+s}) \leq d(h_x, \mu_{r_x+t}) \text{ for all } |s| \leq |t|, \tag{6}$$

as proven in Appendix B. In general, this monotonicity can be achieved, provided that the centroids are arranged directionally according to the ranks, and the instance $h_x$ is located near the centroid $\mu_{r_x}$, as illustrated in Figure 3(a).

To design an embedding space in which instances and centroids are well aligned according to the desideratum in (4), we optimize the encoder parameters by minimizing the loss function

$$\ell_{\text{total}} = \sum_{x \in \mathcal{X}} \ell_{\text{disc}}(x) + \sum_{x,y \in \mathcal{X}} \ell_{\text{order}}(x, y) \tag{7}$$

where $\ell_{\text{disc}}$ is the discriminative loss, and $\ell_{\text{order}}$ is the stochastic order loss.

**Discriminative loss:** To encourage the desideratum in (4), we employ

$$\ell_{\text{disc}}(x) = \sum_{t=1}^{T} \left( D(x, r_x) - D(x, r_x + t) + D(x, r_x) - D(x, r_x - t) \right) \tag{8}$$

$$= \sum_{t=1}^{T} \sum_s (2p_s - p_{s-t} - p_{s+t}) d^2(h_x, \mu_{r_x+s}) \tag{9}$$

$$= \sum_s \alpha_s d^2(h_x, \mu_{r_x+s}) \tag{10}$$

where $\alpha_s = \sum_{t=1}^{T}(2p_s - p_{s-t} - p_{s+t})$.

Note that the coefficient $\alpha_s$ is a discrete approximation of the 2nd-order derivative of the Gaussian distribution, which has inflection points. Therefore, there exists a threshold $\delta$ such that $\alpha_s$ is positive if $|s| < \delta$, while negative otherwise, as shown in Figure 3(b). Hence, to minimize the discriminative loss, $d(h_x, \mu_{r_x+s})$ should be reduced if $|s| < \delta$. In other words, $h_x$ should be attracted to the centroids for the ranks within the range $(r_x - \delta, r_x + \delta)$. On the contrary, if $|s| > \delta$, $d(h_x, \mu_{r_x+s})$ should be increased, thereby repelling $h_x$ from the centroids for the ranks outside $(r_x - \delta, r_x + \delta)$. To summarize, $\ell_{\text{disc}}$ makes each $h_x$ attracted to the corresponding centroid $\mu_{r_x}$ and its neighbors (to consider the label error), but repelled from the other centroids.

**Stochastic order loss**: In order learning (Lim et al., 2020; Lee & Kim, 2021; Lee et al., 2022), pairwise relationships between instances are used to construct a desired embedding space. Thus, while the discriminative loss $\ell_{\text{disc}}$ in (8) considers the geometric configuration of a single instance $x$ with respect to the centroids, the stochastic order loss $\ell_{\text{order}}$ takes into account the geometric configuration of two instances $x$ and $y$ simultaneously.

There are three ordering cases between $x$ and $y$ (Lim et al., 2020):

$$x \prec y \text{ if } \bar{r}_x - \bar{r}_y < -\tau, \quad x \approx y \text{ if } |\bar{r}_x - \bar{r}_y| \leq \tau, \quad x \succ y \text{ if } \bar{r}_x - \bar{r}_y > \tau, \tag{11}$$

where $\tau$ is a threshold. For these three cases, Lee et al. (2022) use margin losses to align instances according to the ranks. Similarly, the proposed $\ell_{\text{order}}$ is based on margin losses. But, unlike Lee et al. (2022), true ranks $\bar{r}_x$ and $\bar{r}_y$ are unknown in SOL. Also, each instance relates to multiple centroids randomly in SOL. We hence develop $\ell_{\text{order}}$ to address these differences.

---

**Algorithm 1** Stochastic Order Learning (SOL)

---

**Input:** A noisy dataset $\mathcal{X}$, $n$ = the number of ranks

1: Initialize centroids $\{\mu_r\}_{r=1}^n$ via (5)
2: **repeat**
3:     Fine-tune the encoder $h$ to minimize $\ell_{\text{total}}$ in (7)          ▷ Network training
4:     **for all** $r = 1, 2, \ldots, n$ **do**
5:         Update centroid $\mu_r$ via (5)          ▷ Centroid rule
6:     **end for**
7:     **for all** $x \in \mathcal{X}$ **do**
8:         Estimate the rank of $x$ via (18)
9:     **end for**
10:   Detect the set of outliers $\bigcup_{r=1}^n \mathcal{X}_r$ via (19)          ▷ Outlier detection
11:   **for all** $x \in \bigcup_{r=1}^n \mathcal{X}_r$ **do**
12:     Estimate the label noise $\hat{e}_x$ via (20)
13:     Refine the label of $x$ via (21)          ▷ Relabeling
14:   **end for**
15: **until** predefined number of epochs

**Output:** Updated labels $\{r_x\}$, centroids $\{\mu_r\}_{r=1}^n$, encoder $h$

---

Since only noisy ranks $r_x$ and $r_y$ are available, the true ranks $\bar{r}_x$ and $\bar{r}_y$ in (11) need to be re-represented using (1). Let the label noise of samples $x$ and $y$ be $s$ and $t$, respectively. Then $\bar{r}_x - \bar{r}_y$ is equal to $r_x - r_y - s + t$. As we model label noise as stochastic variables, we can compute the probabilities for the three ordering cases using (2):

$$\Pr(x \prec y) = \sum_s \sum_{t:r_x-r_y-s+t<-\tau} p_s p_t, \tag{12}$$

$$\Pr(x \approx y) = \sum_s \sum_{t:|r_x-r_y-s+t|\leq\tau} p_s p_t, \tag{13}$$

$$\Pr(x \succ y) = \sum_s \sum_{t:r_x-r_y-s+t>\tau} p_s p_t. \tag{14}$$

Then, we define the margin loss for the case $x \prec y$ as

$$\ell_{x \prec y} = \sum_{r \leq r_x} \max\{D(x,r) - D(y,r) + \gamma, 0\} + \sum_{r \geq r_y} \max\{D(y,r) - D(x,r) + \gamma, 0\} \tag{15}$$

where $\gamma$ is a margin. To minimize the first sum in (15), $D(x,r) - D(y,r) = \sum_s p_s(d^2(h_x, \mu_{r+s}) - d^2(h_y, \mu_{r+s}))$ should be reduced for $r \leq r_x$. Thus, $h_x$ should be near $\mu_{r+s}$, while $h_y$ should be far from $\mu_{r+s}$. Note that this is enforced for small offsets $s$ only because of the Gaussian weights $p_s$. Similarly, for $r \geq r_y$ and a small $s$, $h_x$ should be far from $\mu_{r+s}$, while $h_y$ should be near $\mu_{r+s}$. Hence, $\ell_{x \prec y}$ helps the arrangement of instances and centroids in the embedding space, as illustrated in Figure 3(c). Note that the loss $\ell_{x \succ y}$ for the case $x \succ y$ is formulated in a symmetric manner.

Also, when $x \approx y$, $h_x$ and $h_y$ should be close to each other. We hence define

$$\ell_{x \approx y} = \sum_{r \in \{1,\ldots,n\}} \max(|D(x,r) - D(y,r)| - \gamma, 0). \tag{16}$$

Overall, we define the stochastic order loss as

$$\ell_{\text{order}}(x,y) = \Pr(x \succ y)\ell_{x \succ y} + \Pr(x \approx y)\ell_{x \approx y} + \Pr(x \prec y)\ell_{x \prec y}. \tag{17}$$

### 3.3 INFERENCE RULE

In the testing phase, based on the desideratum in (4), we estimate the rank of an unseen instance $x$ by

$$\hat{r}_x = \arg\min_{r \in \{1,\ldots,n\}} D(x,r). \tag{18}$$

Table 1: Performance comparison on the MORPH II dataset.

| Algorithm | $\kappa = 0.2$ | | $\kappa = 0.3$ | | $\kappa = 0.4$ | | $\kappa = 0.5$ | |
| --- | --- | --- | --- | --- | --- | --- | --- | --- |
| | MAE($\downarrow$) | CS($\uparrow$) | MAE($\downarrow$) | CS($\uparrow$) | MAE($\downarrow$) | CS($\uparrow$) | MAE($\downarrow$) | CS($\uparrow$) |
| SPR (Wang et al., 2022) | 8.446 | 41.71 | 8.881 | 34.79 | 9.239 | 36.89 | 9.993 | 28.14 |
| ACL (Ye et al., 2023) | 9.017 | 36.75 | 9.492 | 35.61 | 9.314 | 35.74 | 9.743 | 34.16 |
| C-Mixup (Yao et al., 2022) | 3.063 | 82.26 | 3.393 | 77.21 | 3.395 | 76.84 | 3.415 | 76.65 |
| POE (Li et al., 2021) | 2.989 | 82.88 | 3.093 | 80.33 | 3.253 | 79.23 | 3.580 | 74.41 |
| MWR (Shin et al., 2022) | 2.570 | 90.07 | 2.693 | 89.25 | 2.851 | 87.16 | 3.089 | 84.34 |
| GOL (Lee et al., 2022) | 2.516 | 90.89 | 2.671 | 89.07 | 2.861 | 85.97 | 3.078 | 84.70 |
| SOL w/o refinement | 2.507 | 91.26 | **2.657** | 88.71 | 2.842 | 86.89 | 2.995 | 84.79 |
| SOL | **2.489** | **91.35** | 2.663 | **89.62** | **2.826** | **87.70** | **2.986** | **85.88** |

Table 2: Performance comparison on the CLAP2015 dataset.

| Algorithm | $\kappa = 0.2$ | | $\kappa = 0.3$ | | $\kappa = 0.4$ | | $\kappa = 0.5$ | |
| --- | --- | --- | --- | --- | --- | --- | --- | --- |
| | MAE($\downarrow$) | CS($\uparrow$) | MAE($\downarrow$) | CS($\uparrow$) | MAE($\downarrow$) | CS($\uparrow$) | MAE($\downarrow$) | CS($\uparrow$) |
| POE (Li et al., 2021) | 4.052 | 70.34 | 4.169 | 68.86 | 4.390 | 65.52 | 4.538 | 65.34 |
| MWR (Shin et al., 2022) | 3.577 | **79.80** | 3.830 | 76.18 | 4.299 | 72.85 | 4.600 | 69.60 |
| GOL (Lee et al., 2022) | 3.624 | 77.94 | 3.866 | 76.03 | 4.105 | 72.10 | 4.284 | 70.90 |
| SOL w/o refinement | **3.556** | 78.41 | 3.766 | 76.37 | 4.058 | 73.68 | 4.208 | **72.57** |
| SOL | 3.559 | 78.68 | **3.764** | **77.11** | **4.002** | 73.68 | **4.170** | 71.64 |

## 3.4 OUTLIER DETECTION AND RELABELING

To obtain a more reliable rank estimator, we identify outliers, likely to have extreme label errors, among instances in the noisy training set and refine their labels by estimating the errors. Then, in turn, we fine-tune the encoder or equivalently revamp the embedding space, so the instances are better arranged based on the refined rank information.

**Outlier detection:** We first estimate the rank of each training instance $x$ using the inference rule in (18). Then, for each rank $r \in \{1, \ldots, n\}$, we detect the set $\mathcal{X}_r$ of outliers by

$$\mathcal{X}_r = \{x : r_x = r \text{ and } |r_x - \hat{r}_x| \geq \beta \cdot \max_{y:r_y=r} |r_y - \hat{r}_y|\} \tag{19}$$

where $\beta \in (0, 1)$ is a constant to control the precision of the outlier detection.

**Relabeling:** For each detected outlier $x \in \bigcup_{r=1}^{n} \mathcal{X}_r$, we estimate its label error as

$$\hat{e}_x = \begin{cases} \frac{1}{2|\mathcal{X}|} \sum_{y \in \mathcal{X}} |r_y - \hat{r}_y| & \text{if } r_x > \hat{r}_x, \\ -\frac{1}{2|\mathcal{X}|} \sum_{y \in \mathcal{X}} |r_y - \hat{r}_y| & \text{if } r_x < \hat{r}_x. \end{cases} \tag{20}$$

Then, from (1), we refine the rank of $x$ by

$$r_x \leftarrow r_x - \hat{e}_x. \tag{21}$$

We note that, in (20), $|\hat{e}_x|$ is determined as half of the average absolute difference between noisy and estimated ranks over all training instances. It is to prevent drastic changes in rank labels, which may rather increase the label errors after relabeling. We repeat the encoder fine-tuning and the outlier detection and relabeling alternately to gradually reduce the label errors and construct a better embedding space. Algorithm 1 summarizes the overall process of SOL.

## 4 EXPERIMENTAL RESULTS

### 4.1 IMPLEMENTATION

We adopt VGG16 (Simonyan & Zisserman, 2015), initialized with the pre-trained parameters on ILSVRC2012 (Deng et al., 2009), as the encoder $h$. We use the Adam optimizer (Kingma & Ba,

Table 3: Performance comparison on the AADB dataset.

| Algorithm | $\kappa = 0.2$ | | $\kappa = 0.3$ | | $\kappa = 0.4$ | | $\kappa = 0.5$ | |
|---|---|---|---|---|---|---|---|---|
| | MAE($\downarrow$) | CS($\uparrow$) | MAE($\downarrow$) | CS($\uparrow$) | MAE($\downarrow$) | CS($\uparrow$) | MAE($\downarrow$) | CS($\uparrow$) |
| POE (Li et al., 2021) | 0.122 | 89.00 | 0.123 | 89.30 | 0.120 | 89.10 | 0.124 | 88.70 |
| MWR (Shin et al., 2022) | 0.123 | 89.00 | 0.124 | 87.60 | 0.122 | 89.80 | 0.131 | 87.70 |
| GOL (Lee et al., 2022) | 0.114 | 92.40 | 0.117 | 91.80 | 0.119 | 91.00 | 0.117 | 92.00 |
| SOL w/o refinement | 0.112 | **93.20** | 0.115 | 93.10 | 0.117 | 91.40 | 0.117 | 92.20 |
| SOL | **0.111** | 92.70 | **0.114** | **93.20** | **0.115** | **92.00** | **0.114** | **92.60** |

Table 4: Performance comparison on the RSNA dataset.

| Algorithm | $\kappa = 0.1$ | | $\kappa = 0.15$ | | $\kappa = 0.2$ | |
|---|---|---|---|---|---|---|
| | MAE($\downarrow$) | CS($\uparrow$) | MAE($\downarrow$) | CS($\uparrow$) | MAE($\downarrow$) | CS($\uparrow$) |
| POE (Li et al., 2021) | 8.517 | 33.50 | 8.614 | 39.50 | 8.796 | 36.00 |
| MWR (Shin et al., 2022) | 7.833 | 41.00 | 8.239 | 39.50 | 8.353 | 39.50 |
| GOL (Lee et al., 2022) | 8.170 | 38.50 | 7.995 | 38.50 | 8.334 | 40.50 |
| SOL w/o refinement | 7.967 | 38.50 | 7.800 | **42.50** | 8.196 | **44.00** |
| SOL | **7.579** | **46.50** | **7.706** | 39.00 | **8.051** | 39.50 |

2015) with a batch size of 32 and a weight decay of $5 \times 10^{-4}$. Also, we set the learning rate to $10^{-4}$ for training on facial age estimation datasets and $5 \times 10^{-5}$ for others. For data augmentation, we do random horizontal flips and random crops. We set $T = 1$ in (8) and $\tau = 3$ in (12)$\sim$(14) as the default mode. More implementation details are in Appendix C.2.

## 4.2 DATASETS

We conduct our experiments on facial age estimation (Ricanek & Tesafaye, 2006; Escalera et al., 2015), aesthetic score regression (Kong et al., 2016), and medical assessment (Halabi et al., 2019) datasets to assess the proposed algorithm. Due to the space limitation, we provide more details of the datasets in Appendix E.

## 4.3 RANK ESTIMATION

We evaluate the rank estimation performances at various severities of label noise. Specifically, we randomly generate label errors according to the zero-mean Gaussian distribution with a standard deviation of $\sigma$, given by

$$\sigma = \kappa \cdot \sigma_{\mathcal{X}} \tag{22}$$

where $\kappa$ is a ratio in $(0, 1)$ to control the severity of label noise, and $\sigma_{\mathcal{X}}$ is the standard deviation of true rank labels in the training set. We list the values of $\sigma$ used for generating label noise in each experiment in Appendix C.3.

**Metrics:** We adopt the mean absolute error (MAE) and cumulative score (CS) metrics. MAE is the average absolute error between estimated and ground-truth ranks, and CS computes the percentage of instances whose absolute errors are less than or equal to a tolerance value of 5.

**Age estimation:** For facial age estimation, we employ MORPH II and CLAP2015, which are two of the most popular datasets. We compare the proposed SOL algorithm with recent noise-robust classification methods SPR (Wang et al., 2022) and ACL (Ye et al., 2023), and noise-robust regression method C-mixup (Yao et al., 2022). Also, we compare the proposed SOL algorithm with POE (Li et al., 2021), MWR (Shin et al., 2022), and GOL (Lee et al., 2022), which are state-of-the-art rank estimators. For fair comparison, the same backbone of VGG16 (Simonyan & Zisserman, 2015) was used for all methods.

Table 1 compares the results on the MORPH II dataset. SPR (Wang et al., 2022) and ACL (Ye et al., 2023), which are recent noise robust-classification methods, do not take ordinal relations of different

Table 5: Comparison of the average noise levels before and after the label refinement.

| | CLAP2015 | | | | | | MORPH II | | | | | |
| --- | --- | --- | --- | --- | --- | --- | --- | --- | --- | --- | --- | --- |
| | MAE | | | Standard Deviation | | | MAE | | | Standard Deviation | | |
| $\kappa = 0.2$ | 1.961 | $\rightarrow$ | 1.959 | 1.508 | $\rightarrow$ | 1.537 | 1.737 | $\rightarrow$ | 1.718 | 1.361 | $\rightarrow$ | 1.343 |
| $\kappa = 0.3$ | 2.970 | $\rightarrow$ | 2.896 | 2.262 | $\rightarrow$ | 2.254 | 2.599 | $\rightarrow$ | 2.534 | 1.991 | $\rightarrow$ | 1.942 |
| $\kappa = 0.4$ | 4.006 | $\rightarrow$ | 3.793 | 3.038 | $\rightarrow$ | 2.899 | 3.504 | $\rightarrow$ | 3.401 | 2.638 | $\rightarrow$ | 2.499 |

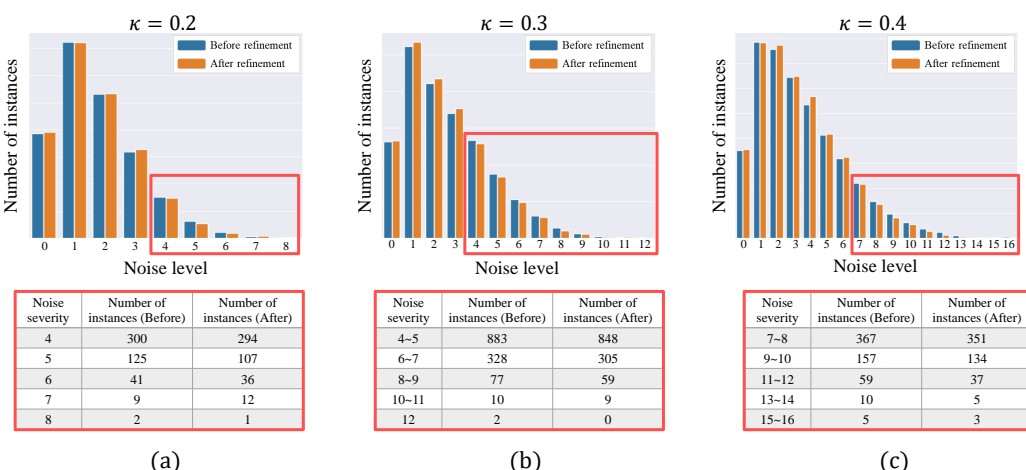

(a)                              (b)                              (c)

Figure 4: Comparison of the numbers of instances at each noise level before and after the label refinement on the MORPH II dataset.

classes into account. Thus, compared to rank estimation methods, they yield poor performances as they cannot distinguish between subtle differences across ordered ranks. To show the effectiveness of our proposed refinement (*i.e.* outlier detection and relabeling) scheme, we list the results of the proposed SOL with and without the refinement scheme. Note that even without the refinement, SOL outperforms the conventional algorithms in terms of MAE at all $\kappa$'s. By further applying the refinement scheme, we reduce extreme label errors in MORPH II, as illustrated in Figure 4. Consequently, the refined labels enable SOL to provide even better rank estimation results.

Table 2 lists the performances on the CLAP dataset. The proposed algorithm also achieves the best MAE scores in all settings. Detecting outliers is more challenging when the average noise level (*i.e.* standard deviation) is relatively low. Thus, at $\kappa = 0.2$, the label refinement rather degrades the MAE, though negligibly. In contrast, for all the other noise settings, SOL achieves the best scores with the refinement. Note that GOL (Lee et al., 2022) also aims to sort instances according to their ranks in the embedding space. Compared to GOL, the proposed SOL achieves a larger CS score gap when the noise level is high ($\kappa = 0.4$) than when it is low ($\kappa = 0.2$). This indicates that, despite of label errors, SOL arranges the instances according to their true ranks more reliably than GOL. In other words, SOL is more noise-robust than GOL.

**Aesthetic score regression:** Table 3 lists the results on the AADB dataset. Note that each CS score in this table computes the percentage of images whose absolute errors are less than or equal to 0.25, rather than 5, because the ranks in AADB are decimal values within $[0, 1]$. It is challenging to estimate aesthetic scores reliably due to the subjectivity and ambiguity of aesthetic criteria. However, SOL performs the best in all tests. It is worth pointing out that, even at the highest $\kappa = 0.5$, the proposed SOL achieves better scores than the second-best GOL at the lowest $\kappa = 0.2$.

**Medical assessment:** In Table 4, we compare the rank estimation results on the RSNA dataset. The proposed SOL yields the best results with no exception. Specifically, at $\kappa = 0.2$, SOL without the refinement scheme outperforms the second-best GOL with significant gaps of 0.138 and 3.5 in the MAE and CS metrics, respectively. It is meaningful because obtaining error-free annotations on medical datasets is difficult and costly in general.

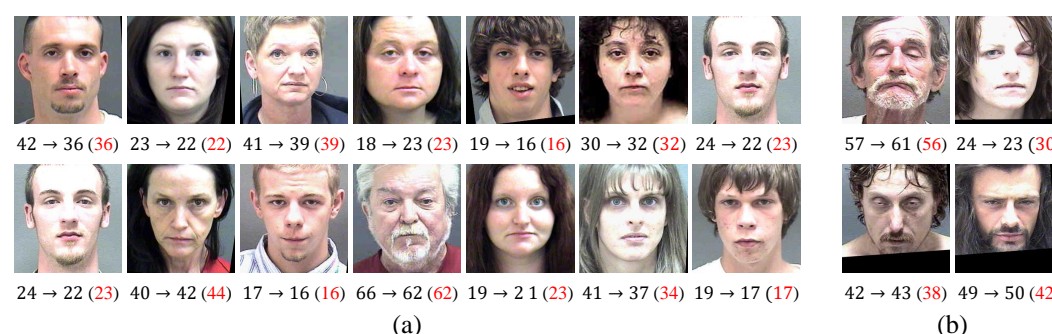

42 → 36 (36)  23 → 22 (22)  41 → 39 (39)  18 → 23 (23)  19 → 16 (16)  30 → 32 (32)  24 → 22 (23)     57 → 61 (56)  24 → 23 (30)

24 → 22 (23)  40 → 42 (44)  17 → 16 (16)  66 → 62 (62)  19 → 2 1 (23)  41 → 37 (34)  19 → 17 (17)     42 → 43 (38)  49 → 50 (42)

(a)                                                                                          (b)

Figure 5: (a) Success and (b) failure cases of the label refinement. Under each image, the noisy, refined, and true ranks are specified: noisy → refined (true).

Table 6: Ablation studies for the loss functions in (7) on the CLAP2015 dataset.

| Method | $\ell_{\text{disc}}$ | $\ell_{\text{order}}$ | $\kappa = 0.2$ | | $\kappa = 0.3$ | | $\kappa = 0.4$ | |
|---|---|---|---|---|---|---|---|---|
| | | | MAE ($\downarrow$) | CS ($\uparrow$) | MAE ($\downarrow$) | CS ($\uparrow$) | MAE ($\downarrow$) | CS ($\uparrow$) |
| I | ✓ | | 15.885 | 23.26 | 16.576 | 19.33 | 38.391 | 17.79 |
| II | | ✓ | 3.598 | 78.13 | 3.888 | 75.90 | 4.038 | 74.70 |
| III | ✓ | ✓ | 3.556 | 78.41 | 3.766 | 76.37 | 4.058 | 73.68 |

## 4.4 ANALYSIS

**Label refinement:** The proposed SOL algorithm is capable of refining noisy ranks, as well as obtaining a reliable rank estimator from a noisy training set. In Table 5, we report MAEs between a noisy rank $r_x$ and the true rank $\bar{r}_x$ and the standard deviations of such noise levels before and after the label refinement. In this test, we use the CLAP2015 and MORPH II datasets. Note that the MAE or the standard deviation is reduced in 11 out of 12 tests, confirming the effectiveness of the label refinement scheme.

For further analysis, we test how the refinement changes the number of instances at each noise level (*i.e.* label error). Figure 4 plots such statistics on the MORPH dataset at various $\kappa$'s. The red boxes in Figure 4 specify the numbers of instances with high noise levels. We see that the number of instances with an extreme noise level is reduced in general. Especially, at $\kappa = 0.4$, the number of instances with $2 \leq e_x \leq 4$ is increased, while that with $e_x \geq 7$ is reduced significantly. It is desirable because severe label errors hinder the construction of a well-sorted embedding space. Consequently, in Tables 1∼4, the refinement generally boosts the performances of SOL.

We also provide examples of detected outliers in Figure 5. These examples are from MORPH II at $\kappa = 0.4$. With the refinement scheme, instances with extreme label errors in Figure 5(a) are relabelled more faithfully to the true ranks. Along with these successful cases, we also show some failures cases in Figure 5(b). Although SOL succeeds in detecting these instances as outliers, it fails to correct the labels in the right directions. We see that facial images with closed eyes tend to be corrected wrongly. More label refinement results are provided in Appendix D.1.

**Ablation study:** Table 6 compares ablated methods for the loss functions in (7). Method I employs the discriminative loss $\ell_{\text{disc}}$ only, while method II does the stochastic order loss $\ell_{\text{order}}$ only. Compared with method III (SOL), both methods I and II degrade the rank estimation results, indicating that both losses contribute to the performance improvement and are complementary to each other. Note that method I yields poor results, since the discriminative loss alone cannot construct a meaningful embedding space; it is trivial to reduce $\ell_{\text{disc}}$ to zero by merging all instances into a single point in the embedding space. However, by comparing methods II and III, we see that $\ell_{\text{disc}}$ helps to sort instances in the embedding space by properly attracting and repelling instances according to their ranks.

Table 7: Comparison of rank estimation results on the MORPH II dataset according to $\beta$ in (19).

| | $\beta = 0.8$ | | $\beta = 0.85$ | | $\beta = 0.9$ | | $\beta = 0.95$ | |
| --- | --- | --- | --- | --- | --- | --- | --- | --- |
| | MAE ($\downarrow$) | CS ($\uparrow$) | MAE ($\downarrow$) | CS ($\uparrow$) | MAE ($\downarrow$) | CS ($\uparrow$) | MAE ($\downarrow$) | CS ($\uparrow$) |
| $\kappa = 0.2$ | 2.511 | 90.07 | 2.506 | 90.80 | 2.489 | 91.35 | 2.525 | 90.80 |
| $\kappa = 0.3$ | 2.672 | 90.16 | 2.715 | 88.89 | 2.663 | 89.62 | 2.677 | 89.80 |
| $\kappa = 0.4$ | 2.839 | 87.25 | 2.811 | 87.61 | 2.826 | 87.70 | 2.897 | 86.98 |

Table 8: Comparison of rank estimation results on the MORPH II dataset according to $T$ in (8).

| | $T = 1$ | | $T = 2$ | | $T = 3$ | |
| --- | --- | --- | --- | --- | --- | --- |
| | MAE ($\downarrow$) | CS ($\uparrow$) | MAE ($\downarrow$) | CS ($\uparrow$) | MAE ($\downarrow$) | CS ($\uparrow$) |
| $\kappa = 0.2$ | 2.507 | 91.26 | 2.573 | 90.53 | 2.565 | 90.80 |
| $\kappa = 0.3$ | 2.657 | 88.71 | 2.685 | 89.71 | 2.658 | 89.53 |
| $\kappa = 0.4$ | 2.842 | 86.89 | 2.839 | 87.34 | 2.810 | 87.52 |

Table 9: Comparison of rank estimation results on the MORPH II dataset according to $\sigma$ in (2).

| | $\sigma = 0.5$ | | $\sigma = 1$ | | $\sigma = 1.5$ | |
| --- | --- | --- | --- | --- | --- | --- |
| | MAE ($\downarrow$) | CS ($\uparrow$) | MAE ($\downarrow$) | CS ($\uparrow$) | MAE ($\downarrow$) | CS ($\uparrow$) |
| $\kappa = 0.2$ | 2.557 | 90.62 | 2.507 | 91.26 | 2.529 | 90.80 |
| $\kappa = 0.3$ | 2.682 | 88.62 | 2.657 | 88.71 | 2.689 | 89.44 |
| $\kappa = 0.4$ | 2.850 | 86.98 | 2.842 | 86.89 | 2.869 | 86.70 |

**Analysis on $\beta$:** Table 7 compares the MAE and CS scores at different $\beta$'s on the MORPH II dataset. Note that $\beta$ is a parameter to control the precision of outlier detection in (19). In general, at high $\beta \geq 0.9$, SOL yields better results than at low $\beta < 0.9$. It is because inaccurate outlier detection at low $\beta$ may deteriorate the network training by increasing the label noise. Thus, we set $\beta = 0.9$ as the default option.

**Analysis on $T$:** Table 8 compares the results at different $T$'s on MORPH II. $T$ is a hyper-parameter used in (8). Given an instance, using a bigger $T$ enforces that the stochastic similarities to more neighboring ranks are considered in the optimization. In the case of $\kappa = 0.4$, the performance gets better with a bigger $T$. However, excluding this case, using $T = 1$ yields decent results. Thus, we set $T = 1$ in the default mode.

**Analysis on $\sigma$:** Table 9 compares the MAE and CS scores at different $\sigma$'s on MORPH II. In (2), $\sigma$ is the standard deviation of label errors $\mathbf{e}$. The larger $\sigma$, the more strongly each instance $x$ is associated with the centroids for distant ranks from $r_x$. Therefore, it may hinder constructing a rank discriminative embedding space. On the other hand, if $\sigma$ is small, the network training is vulnerable to label errors because each instance $x$ is related to nearby centroids only. Hence, at $\sigma = 1$, which is the default option, SOL achieves good results in most tests.

## 5 CONCLUSIONS

The SOL algorithm for rank estimation in the presence of label noise was proposed in this work. First, we represented label errors as random variables. Then, we formulated a desideratum to reduce the dissimilarity of an instance from the stochastically related centroids. Using the discriminative loss and the stochastic order loss, we constructed an embedding space satisfying the desideratum. Specifically, we trained a network to arrange the instances according to their unknown true ranks. Moreover, by using the trained network, we identified outliers, likely to have extreme label errors, and relabelled them for data refinement. Extensive experiments on various rank estimation tasks demonstrated that the proposed SOL algorithm yields decent rank estimation results even when labels are corrupted by noise.

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

## A    DERIVATION OF CENTROID RULE IN (5)

Based on the desideratum in (4), we formulate a cost function

$$
\begin{align}
J &= \sum_{x \in \mathcal{X}} D(x, r_x) \tag{23} \\
&= \sum_{x \in \mathcal{X}} \sum_s p_s d^2(h_x, \mu_{r_x+s}) \tag{24} \\
&= \sum_{x \in \mathcal{X}} \sum_s p_s (\mu_{r_x+s}^T \mu_{r_x+s} - 2h_x^T \mu_{r_x+s} + h_x^T h_x) \tag{25} \\
&= \sum_{x \in \mathcal{X}} \sum_r p_{r-r_x} (\mu_r^T \mu_r - 2h_x^T \mu_r + h_x^T h_x). \tag{26}
\end{align}
$$

We then update the centroids $\{\mu_r\}_{r=1}^n$ to minimize the cost function $J$. By differentiating $J$ with respect to each $\mu_r$ and setting it to zero, we have

$$
\frac{\partial J}{\partial \mu_r} = \sum_{x \in \mathcal{X}} p_{r-r_x}(2\mu_r - 2h_x) = 0. \tag{27}
$$

Hence, the optimal centroid is given by

$$
\mu_r = \frac{\sum_{x \in \mathcal{X}} p_{r-r_x} h_x}{\sum_{x \in \mathcal{X}} p_{r-r_x}}, \qquad r \in \{1, \ldots, n\}. \tag{28}
$$

## B    DERIVATION OF MONOTONICITY CONSTRAINT IN (6)

The desideratum in (4) can be written as

$$
\sum_s p_s d^2(h_x, \mu_{r_x+s}) \leq \sum_s p_s d^2(h_x, \mu_{(r_x+k)+s}) \quad \text{for all } k. \tag{29}
$$

For simpler notations, let $L_s \triangleq d^2(h_x, \mu_{r_x+s})$. Then, the desideratum is given by

$$
\sum_s p_s L_s \leq \sum_s p_s L_{s+k} \quad \text{for all } k. \tag{30}
$$

First, let us consider the case for $k = 1$. From (30), we have

$$
\begin{align}
\cdots + p_2 L_{-2} + p_1 L_{-1} + p_0 L_0 + p_1 L_1 + p_2 L_2 + \cdots &\leq \tag{31} \\
\cdots + p_3 L_{-2} + p_2 L_{-1} + p_1 L_0 + p_0 L_1 + p_1 L_2 + \cdots &
\end{align}
$$

since $p_s$ in (2) is symmetric. Thus,

$$
(p_0 - p_1)(L_0 - L_1) + (p_1 - p_2)(L_{-1} - L_2) + (p_2 - p_3)(L_{-2} - L_3) + \cdots \leq 0. \tag{32}
$$

Because $p_s$ in (2) is also unimodal, the coefficients $(p_s - p_{s+1})$ are positive for all $s \geq 0$. Hence, the inequality in (32) is satisfied if

$$
L_0 \leq L_1, \quad L_{-1} \leq L_2, \quad L_{-2} \leq L_3, \quad \cdots \tag{33}
$$

or equivalently

$$
L_{-m} \leq L_{1+m} \quad \text{for all } m \geq 0. \tag{34}
$$

Next, let us consider the case for $k = 2$. Similar to (32), we have

$$
(p_0 - p_2)(L_0 - L_2) + (p_1 - p_3)(L_{-1} - L_3) + (p_2 - p_4)(L_{-2} - L_4) + \cdots \leq 0. \tag{35}
$$

This is satisfied if

$$
L_{1-m} \leq L_{1+m} \quad \text{for all } m \geq 0. \tag{36}
$$

In general, if $k \geq 1$, we have the following condition:

$$
L_{\lfloor \frac{k}{2} \rfloor - m} \leq L_{\lceil \frac{k}{2} \rceil + m} \quad \text{for all } m \geq 0. \tag{37}
$$

Note that (34) and (36) are special cases of (37). Symmetrically, if $k \leq -1$, we have the condition:

$$
L_{\lfloor \frac{k}{2} \rfloor - m} \geq L_{\lceil \frac{k}{2} \rceil + m} \quad \text{for all } m \geq 0. \tag{38}
$$

Both conditions in (37) and (38) are satisfied if

$$
L_0 \leq L_{\pm 1} \leq L_{\pm 2} \leq L_{\pm 3} \leq \cdots, \tag{39}
$$

implying that $L_k$ should be a monotonic increasing function of $|k|$. Rewriting this monotonicity constraint in the original notations, we have the sufficient condition in (6),

$$
d(h_x, \mu_{r_x+s}) \leq d(h_x, \mu_{r_x+t}) \quad \text{for all } |s| \leq |t|. \tag{40}
$$

# C IMPLEMENTATION DETAILS

## C.1 NETWORK ARCHITECTURE

As described in Section 3.2, we employ an encoder to map each instance into a feature vector in an embedding space. The network structure used for the encoder $h$ is specified in Figure 6. The encoder is based on the VGG16 network and takes a $224 \times 224 \times 3$ image as input.

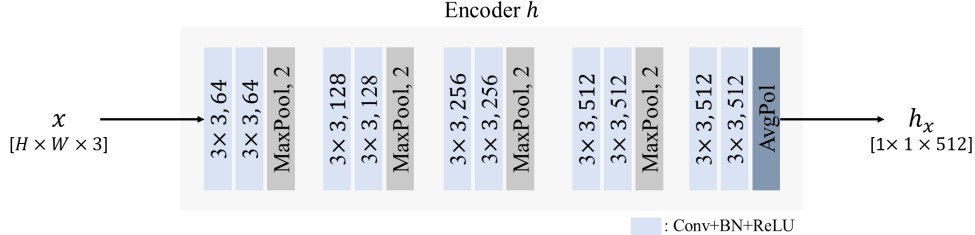

Figure 6: Network structure of the encoder $h$.

## C.2 HYPER-PARAMETER SETTINGS

Using an encoder based on the VGG16 network, we train the network for 100 epochs. According to the dataset, some of the hyper-parameters such the learning-rate, $\beta$ and $\tau$ are set differently. Table 10 summarizes the hyper-parameters used for training and detecting the outliers.

Table 10: Hyper-parameter settings

| Dataset | Learning rate | Batch size | $\beta$ | $\tau$ | $T$ | $\gamma$ | $\sigma$ |
|---------|---------------|------------|---------|--------|-----|----------|----------|
| MORPH II | $10^{-4}$ | 32 | 0.9 | 3 | 1 | 0.25 | 1 |
| CLAP2015 | $10^{-4}$ | 32 | 0.85 | 3 | 1 | 0.25 | 1 |
| AADB | $5 \times 10^{-5}$ | 32 | 0.85 | 5 | 1 | 0.25 | 1 |
| RSNA | $5 \times 10^{-5}$ | 32 | 0.9 | 3 | 1 | 0.25 | 1 |

## C.3 DETAILS OF $\sigma$ IN (22)

We specify the actual values of $\sigma$ used for generating the Gaussian noise in (22). Table 11 lists the values of $\sigma$ according to $\kappa$.

Table 11: Actual values of $\sigma$ according to $\kappa$.

| | $\sigma$ | | | | | |
|---|---|---|---|---|---|---|
| | $\kappa = 0.1$ | $\kappa = 0.15$ | $\kappa = 0.2$ | $\kappa = 0.3$ | $\kappa = 0.4$ | $\kappa = 0.5$ |
| MORPH II | 1.092 | 1.638 | 2.184 | 3.276 | 4.368 | 5.460 |
| CLAP2015 | 1.235 | 1.853 | 2.471 | 3.706 | 4.941 | 6.177 |
| AADB | 0.018 | 0.0276 | 0.037 | 0.055 | 0.074 | 0.102 |
| RSNA | 4.118 | 6.177 | 8.326 | 12.355 | 16.473 | 20.591 |

# D   MORE RESULTS

## D.1   MORE EXAMPLES OF DETECTED OUTLIERS

We provide more examples of the detected outlier instances. Figures 7, 8, and 9 show results from the MORPH II, CLAP2015, and AADB dataset, respectively

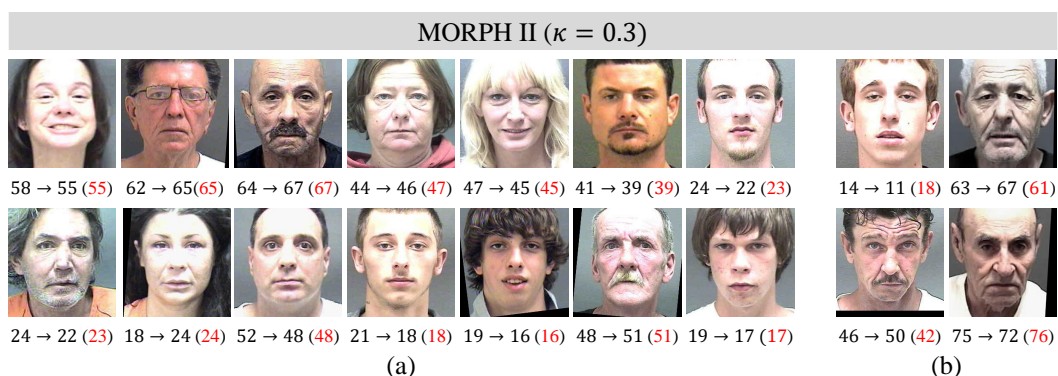

Figure 7: (a) Success and (b) failure cases of the label refinement on the MORPH II dataset at $\kappa = 0.3$. Under each image, the noisy, refined, and true ranks are specfied: noisy $\rightarrow$ refined (true).

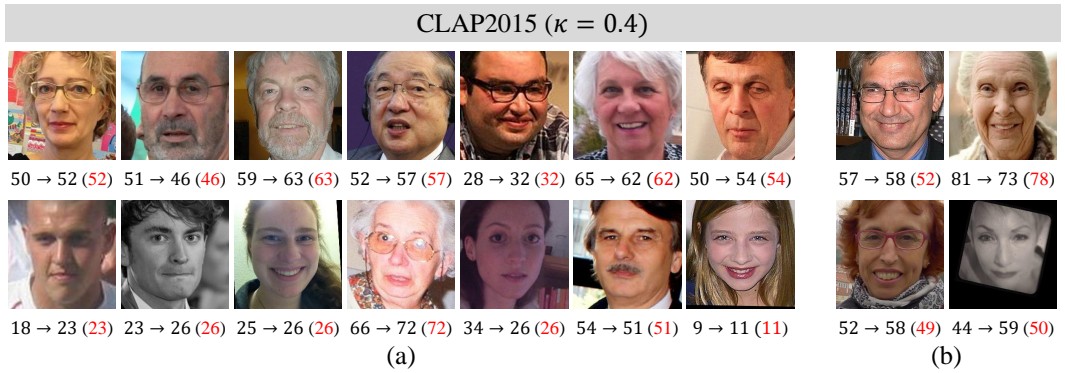

Figure 8: (a) Success and (b) failure cases of the label refinement on the CLAP dataset at $\kappa = 0.4$. Under each image, the noisy, refined, and true ranks are specfied: noisy $\rightarrow$ refined (true).

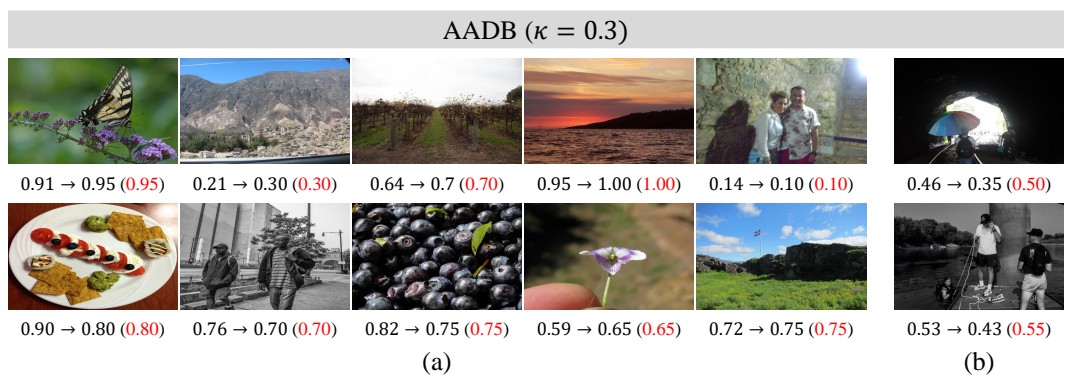

Figure 9: (a) Success and (b) failure cases of the label refinement on the AADB dataset at $\kappa = 0.3$. Under each image, the noisy, refined, and true ranks are specfied: noisy $\rightarrow$ refined (true).

# E DATASETS

**MORPH II** (Ricanek & Tesafaye, 2006)**:** It is a dataset for facial age estimation, consisting of 55K facial images in the age range $[16, 77]$. It provides age, gender, and race labels. As in Chang et al. (2011), we use 5,492 Caucasian images, which are divided into training and test sets with ratio 8:2.

**CLAP2015** (Escalera et al., 2015)**:** It is for apparent age estimation; the apparent age of each image was rated by at least 10 annotators within the range $[3, 85]$, and the mean rating was used as the ground-truth. This dataset provides 4,691 facial images in total, which are split into 2,476 for training, 1,136 for validation, and 1,079 for testing.

**AADB** (Kong et al., 2016)**:** It is a dataset for aesthetic score regression, composed of 10,000 photographs of various themes such as scenery and close-up. We use 8,500 images for training, 500 for validation, and 1,000 for testing. Each image is annotated with an aesthetic score in $[0, 1]$. We quantize the continuous score with a step size of 0.01 to have 101 score ranks.

**RSNA** (Halabi et al., 2019)**:** It is a dataset for pediatric bone age assessment, containing 14,236 hand radiographs. We employ the official evaluation protocol in Halabi et al. (2019): 12,611 for training, 1,425 for validation, and 200 for testing. The bone age range is $[0, 216]$ in months.

# F MORE EXPERIMENTS

## F.1 MORE COMPARISONS

Due to limited space, in the main paper, we compare the proposed algorithm with only rank estimation methods for CLAP2015 (Escalera et al., 2015), AADB (Kong et al., 2016), and RSNA (Halabi et al., 2019) datasets. Here, we include the results of conventional noise-robust classification and regression methods.

Table 12 is an extended version of Table 2.

Table 12: Performance comparison on the CLAP2015 dataset.

| Algorithm | $\kappa = 0.2$ | | $\kappa = 0.3$ | | $\kappa = 0.4$ | | $\kappa = 0.5$ | |
|---|---|---|---|---|---|---|---|---|
| | MAE($\downarrow$) | CS($\uparrow$) | MAE($\downarrow$) | CS($\uparrow$) | MAE($\downarrow$) | CS($\uparrow$) | MAE($\downarrow$) | CS($\uparrow$) |
| SPR (Wang et al., 2022) | 9.170 | 44.21 | 9.215 | 43.19 | 9.534 | 40.12 | 9.832 | 37.72 |
| ACL (Ye et al., 2023) | 9.483 | 41.06 | 9.239 | 39.57 | 9.583 | 452.3 | 9.651 | 38.14 |
| C-Mixup (Yao et al., 2022) | 5.042 | 61.65 | 5.285 | 58.71 | 5.302 | 58.52 | 5.576 | 55.40 |
| POE (Li et al., 2021) | 4.052 | 70.34 | 4.169 | 68.86 | 4.390 | 65.52 | 4.538 | 65.34 |
| MWR (Shin et al., 2022) | 3.577 | **79.80** | 3.830 | 76.18 | 4.299 | 72.85 | 4.600 | 69.60 |
| GOL (Lee et al., 2022) | 3.624 | 77.94 | 3.866 | 76.03 | 4.105 | 72.10 | 4.284 | 70.90 |
| SOL w/o refinement | **3.556** | 78.41 | 3.766 | 76.37 | 4.058 | **73.68** | 4.208 | **72.57** |
| SOL | 3.559 | 78.68 | **3.764** | **77.11** | **4.002** | **73.68** | **4.170** | 71.64 |

Table 13 is an extended version of Table 3.

Table 13: Performance comparison on the AADB dataset.

| Algorithm | $\kappa = 0.2$ | | $\kappa = 0.3$ | | $\kappa = 0.4$ | | $\kappa = 0.5$ | |
|---|---|---|---|---|---|---|---|---|
| | MAE($\downarrow$) | CS($\uparrow$) | MAE($\downarrow$) | CS($\uparrow$) | MAE($\downarrow$) | CS($\uparrow$) | MAE($\downarrow$) | CS($\uparrow$) |
| SPR (Wang et al., 2022) | 0.149 | 81.20 | 0.150 | 82.10 | 0.151 | 81.60 | 0.158 | 81.10 |
| ACL (Ye et al., 2023) | 0.147 | 82.90 | 0.148 | 82.50 | 0.157 | 79.43 | 0.159 | 78.95 |
| C-Mixup (Yao et al., 2022) | 0.119 | 91.13 | 0.122 | 89.31 | 0.130 | 88.51 | 0.135 | 86.39 |
| POE (Li et al., 2021) | 0.122 | 89.00 | 0.123 | 89.30 | 0.120 | 89.10 | 0.124 | 88.70 |
| MWR (Shin et al., 2022) | 0.123 | 89.00 | 0.124 | 87.60 | 0.122 | 89.80 | 0.131 | 87.70 |
| GOL (Lee et al., 2022) | 0.114 | 92.40 | 0.117 | 91.80 | 0.119 | 91.00 | 0.117 | 92.00 |
| SOL w/o refinement | 0.112 | **93.20** | 0.115 | 93.10 | 0.117 | 91.40 | 0.117 | 92.20 |
| SOL | **0.111** | 92.70 | **0.114** | **93.20** | **0.115** | **92.00** | **0.114** | **92.60** |

Table 14 is an extended version of Table 4.

Table 14: Performance comparison on the RSNA dataset.

| Algorithm | $\kappa = 0.1$ | | $\kappa = 0.15$ | | $\kappa = 0.2$ | |
|---|---|---|---|---|---|---|
| | MAE($\downarrow$) | CS($\uparrow$) | MAE($\downarrow$) | CS($\uparrow$) | MAE($\downarrow$) | CS($\uparrow$) |
| SPR (Wang et al., 2022) | 33.80 | 14.00 | 36.48 | 9.50 | 34.88 | 6.50 |
| ACL (Ye et al., 2023) | 35.09 | 11.33 | 35.15 | 11.25 | 35.26 | 10.17 |
| C-Mixup (Yao et al., 2022) | 8.200 | 40.10 | 8.621 | 35.42 | 9.054 | 33.33 |
| POE (Li et al., 2021) | 8.517 | 33.50 | 8.614 | 39.50 | 8.796 | 36.00 |
| MWR (Shin et al., 2022) | 7.833 | 41.00 | 8.239 | 39.50 | 8.353 | 39.50 |
| GOL (Lee et al., 2022) | 8.170 | 38.50 | 7.995 | 38.50 | 8.334 | 40.50 |
| SOL w/o refinement | 7.967 | 38.50 | 7.800 | 42.50 | 8.196 | 44.00 |
| SOL | 7.579 | 46.50 | 7.706 | 39.00 | 8.051 | 39.50 |

## F.2 PERFORMANCE ON PARTIALLY CORRUPTED DATA

Table 15 lists the MAE results of when only 10% of the total data have the risk of labeling errors. We compare the proposed algorithm to the state-of-the-art algorithm GOL (Lee et al., 2022). Even in this case, the proposed SOL achieves better results than GOL.

Table 15: MAE results of when only 10% of total data is corrupted on the CLAP2015 dataset.

| Algorithm | $\kappa = 0.2$ | $\kappa = 0.3$ | $\kappa = 0.4$ | $\kappa = 0.5$ |
|---|---|---|---|---|
| GOL (Lee et al., 2022) | 3.442 | 3.540 | 3.590 | 3.690 |
| SOL (Proposed) | 3.420 | 3.505 | 3.549 | 3.639 |

## F.3 RELABELING SCHEME

In our proposed relabeling scheme, the rank of detected outliers are adjusted by the same magnitude using (20). Here, we assess the performance of when each detected outlier is relabeled using different magnitudes. Specifically, we adjust the rank of each outlier instance by half of the absolute difference between its noisy and estimated rank. Table 16 lists the results on the CLAP2015 dataset. Compared to when no relabeling method is used, method II shows performance improvement. However, our proposed relabeling scheme provides better results. Using the same average values to adjust the ranks prevent drastic changes in rank labels, thus it results in more stable performance.

Table 16: Ablation studies for relabeling scheme on the CLAP2015 dataset at $\kappa = 0.4$.

| | Relabeling mechanism | MAE ($\downarrow$) | CS ($\uparrow$) |
|---|---|---|---|
| I | No relabeling | 4.058 | 73.68 |
| II | Different magnitudes | 4.012 | 72.75 |
| III | Proposed | 4.002 | 73.68 |

## F.4 LIMITATIONS

SOL is the first attempt to extend the concept of order learning to noise-robust rank estimation. Hence, not all details in real-world scenarios are covered. It is a future research issue to set noise distributions adaptively according to each instance, and design an adaptive mechanism for selecting optimal values of hyper-parameters.

## F.5 COMPLEXITY

Table 17 compares the time required to compute the centroids in (5) to the total training time. We use an RTX 4090 GPU to measure the times. Even for the RSNA dataset which consists of 12,611 training samples, it takes only a few minutes to compute the centroids. This is fast enough for most use cases since centroid update is performed only once at each epoch. Thus, the centroid update accounts for only a small portion of the overall training time. Moreover, for even larger datasets, we can reduce the computation time by sub-sampling the training instances for the centroid update.

Table 17: Analysis on the processing time required for training one epoch on the MORPH II, CLAP2015, AADB, and RSNA datasets.

|  | MORPH II | CLAP2015 | AADB | RSNA |
|---|---|---|---|---|
| Centroid computation | 6.10s | 5.08s | 39.18s | 286.06s |
| Training 1 epoch | 60.17s | 52.07s | 145.43s | 1160.66s |

For large scale training, memory efficiency is also important. Hence, we compare the number of parameters of SOL with that of conventional methods. Table 18 lists the results. SOL requires the smallest number of parameters, showing its potential for large scale applications.

Table 18: Comparison with conventional algorithms in terms of network complexity.

| Algorithm | # of parameters |
|---|---|
| ACL (Ye et al., 2023) | 134.68M |
| MWR (Shin et al., 2022) | 139.41M |
| GOL (Lee et al., 2022) | 14.75M |
| SOL | 14.72M |

