# OpenReview forum: "Stochastic Order Learning: An Approach to Rank Estimation Using Noisy Data"
_ICLR.cc/2025/Conference — Submitted to ICLR 2025_

### Official Review · Reviewer_g1N2 · 2024-10-31

**Soundness:** 2
**Presentation:** 1
**Contribution:** 2
**Rating:** 3
**Confidence:** 3

**Summary:**

This paper solves the rank estimation problem under noisy data. The main novelty of this work is handling the input data with label noise, which seems limited, as other works can also handle noise data. Moreover, the motivation of the adopted techniques is also not clear. The experimental evaluation of this paper is good.

**Strengths:**

This paper proposes a model to handle the noisy rank learning problem. The experimental evaluation is good.

**Weaknesses:**

Many descriptions are not clear. For example:
What is the difference between rank estimation and order learning?
In Eq. (1), the ground truth label is represented by consecutive integers, while the observed noisy rank is assumed to be the sum of the true rank with some Gaussian noise. When the Gaussian noise is added, will the observed rank still be integers?
Besides, why does the noise follow a Gaussian distribution?
In line 137, what is the relationship between e and e_x?

**Questions:**

Please see the weakness.
Also, please describe the motivation for the proposed method.

---

> ### Author Response · Authors · 2024-11-15
> **Rebuttal by Authors**
>
> Thank you for your constructive comments. Please find our responses below.
> ***
>
> > **Rank estimation vs. order learning**
>
> Rank estimation is a "task" that aims to predict the ordered class of an object. On the other hand, as stated in L106, **order learning is an approach to rank estimation** problem. Specifically, order learning algorithms exploit ordering relationship between instances for rank estimation.
>
> > **Gaussian noise**
>
> * As stated in L137, the noise is assumed to have a "discrete" Gaussian distribution. Therefore, the observed rank $r_x$ in Eq.(1) will still be integers after the Gaussian noise is added.
>
> * In practice, when annotators make errors in ranking, they are more likely to choose neighboring ranks than far-away ranks. As stated in L140-142, the symmetric($p_s=p_{-s}$), and unimodal($p_s \geq p_t$ for $0\leq s \leq t$) property of a Gaussian noise distribution well-models this trait. Thus, we chose the noise to follow a Gaussian distribution.
>
> * As stated in L136, $e_x$ is the label error of an instance $x$ and $\mathbf{e}$ is the random variable underlying each error $e_x$. We are not sure of what exactly the point of the question here is. We will do our best to answer your question if you could specify it.
>
>
> > **Motivation**
>
> Our motivation was to make a noise-robust algorithm suitable for rank estimation. Although many noise-robust classification methods have been developed, they yield poor results when applied to rank estimation. Most rank estimation methods, on the other hand, do not consider label noise. Specifically, the state-of-the-art method[1] designs an embedding space that arranges instances according to deterministic ranks. Using deterministic ranks fails to consider the possibility of rank label errors. In order to handle noisy data, we were motivated to use stochastic variables to model the label errors. We derived an embedding space in which instances are arranged based on probabilistically associated ranks instead of deterministic ranks.
>
>
> [1] Geometric order learning for rank estimation. In NIPS, 2022.
> ***
> If you have any additional concerns, please let us know. We will address them faithfully. Thank you again for your constructive comments.

---

### Official Review · Reviewer_En6x · 2024-10-31

**Soundness:** 2
**Presentation:** 3
**Contribution:** 2
**Rating:** 5
**Confidence:** 3

**Summary:**

This paper introduces a novel algorithm called Stochastic Order Learning (SOL) for reliable rank estimation in the presence of label noise. The proposed method models label errors as random variables and formulates a desideratum to minimize dissimilarity from stochastically related centroids. By employing discriminative loss and stochastic order loss, the algorithm constructs an embedding space that effectively arranges instances according to their true ranks. Additionally, it incorporates outlier detection and relabeling schemes to refine noisy data, demonstrating robust performance across various benchmark tasks.

**Strengths:**

1. The proposed algorithm in this paper demonstrates reasonable formulation for rank estimation in the presence of label noise, with clear mathematical derivations.


2. The experimental section is adequately designed, covering multiple benchmark datasets to validate the algorithm's performance.

**Weaknesses:**

1. The novelty of the paper is relatively moderate. While it builds upon existing research, it lacks significant innovative contributions.

2. The Dataset Section can be placed in the Appendix to allow more space for detailed analysis of experimental results.


3. The paper lacks comparisons with algorithms from the past two years. Including recent benchmarks would provide a more comprehensive evaluation for the proposed method. Besides, the number of comparison algorithms is too small. Adding more comparison algorithms will further highlight its advantages.

**Questions:**

See Weaknesses

---

> ### Author Response · Authors · 2024-11-15
> **Rebuttal by Authors**
>
> Thank you for your constructive review and valuable suggestions. We have revised the paper to address your comments faithfully, and highlighted the revised parts in blue. Please find our responses below.
> ***
>
> > **Novelty**
>
> SOL is the first order-learning algorithm to handle label noise by designing a stochastic approach. GOL[1] designs an embedding space that arranges instances according to their ranks. However, the ranks assigned are deterministic, failing to take label noise into account. To model label errors, we use stochastic variables and we develop the discriminative loss and the stochastic order loss to derive an embedding space in which instances are arranged based on probabilistically associated ranks. Furthermore, we propose outlier detection and relabeling schemes to reduce the overall noise level of a given dataset. Experiments on various datasets prove that our algorithm shows the best performance for rank estimation tasks under label noise.
>
> [1] Geometric order learning for rank estimation. In NIPS, 2022.
>
> > **More comparisons**
>
> As you suggested, we compare our algorithms with more comparative methods[1,2,3] including recent benchmarks. Below lists the comparisons on the MORPH II dataset. The proposed SOL still achieves the best scores. Recent noise robust-classification methods[1,2] cannot distinguish between subtle differences across ordered ranks, thus they yield poor performances. The results for other three datasets(CLAP2015, AADB, RSNA) are similar as well. We included and discussed these comparisons in the revised paper. Please see Table 1 on page 6 and Tables 12, 13, 14 on page 16 and 17.
>
> |                    | $\kappa=0.2$      |                  | $\kappa=0.3$      |                  | $\kappa=0.4$      |                  | $\kappa=0.5$      |                  |
> |--------------------|:-------------------:|:------------------:|:-------------------:|:------------------:|:-------------------:|:------------------:|:-------------------:|:------------------:|
> | Algorithm          | MAE| CS   | MAE| CS   | MAE| CS  | MAE | CS  |
> | SPR [1]                | 8.446             | 41.71            | 8.881             | 34.79            | 9.239             | 36.89            | 9.993             | 28.14            |
> | ACL [2]               | 9.017             | 36.75            | 9.492             | 35.61            | 9.314             | 35.74            | 9.743             | 34.16            |
> | C-Mixup [3]            | 3.063 | 82.26 | 3.393 | 77.21 | 3.395 | 76.84 | 3.415 | 76.65 |
> | POE                | 2.989             | 82.88            | 3.093             | 80.33            | 3.253             | 79.23            | 3.580             | 74.41            |
> | MWR                | 2.570             | 90.07            | 2.693             |89.25 | 2.851             |87.16| 3.089             | 84.34            |
> | GOL                | 2.516             | 90.89            | 2.671             | 89.07            | 2.861             | 85.97            | 3.078             | 84.70            |
> | SOL w/o refinement |2.507 |91.26 | **2.657**         | 88.71            |2.842| 86.89            |2.995|84.79|
> | SOL                | **2.489**         | **91.35**        |2.663| **89.62**        | **2.826**         | **87.70**        | **2.986**         | **85.88**        |
>
> [1] Scalable penalized regression for noise detection in learning with noisy labels. In CVPR, 2022.
>
> [2] Active negative loss functions for learning with noisy labels. In NIPS, 2023.
>
> [3] C-Mixup: Improving generalization in regression. In NIPS, 2022.
>
> > **Dataset section**
>
> We agree with you that moving the dataset section to the Appendix adds more space for experimental results. Following your suggestion, we moved the detailed dataset descriptions to Appendix E on page 16 of the revised paper. Thank you for your constructive comment.
>
> ***
> We have revised our paper to address your comments faithfully. We hope that this revision resolves your concerns. If you have any additional comments, please let us know.

---

### Official Review · Reviewer_nnr1 · 2024-11-02

**Soundness:** 2
**Presentation:** 2
**Contribution:** 2
**Rating:** 3
**Confidence:** 4

**Summary:**

This paper presents a novel algorithm to address noise in rank ordering within learning-to-rank tasks. Extensive experiments are conducted to demonstrate the effectiveness of the proposed method.

**Strengths:**

This paper addresses a practical, real-world setting, and the experiments effectively demonstrate the method's effectiveness.

**Weaknesses:**

1. The problem of noisy classification is well-studied, particularly in image classification (e.g., [1]). However, the authors do not clearly explain how their approach to learning with rank labels differs from or improves upon standard classification methods.
2. The writing quality requires improvement. Several notations are undefined, and some equations are presented without adequate derivation or verification.


[1] Xiao, Tong, et al. "Learning from massive noisy labeled data for image classification." Proceedings of the IEEE conference on computer vision and pattern recognition. 2015.

**Questions:**

1. Where is the definition of $d(\cdot, \cdot)$ in Eq.(3)
2. How do the authors ensure that the noise $e_x$ remains within the range of true labels in Equations Eq.(1) and Eq.(2)
3. In line 155, could the authors clarify why $\mu_{r_{x + s}}$ the probablity $p_s$?
4. According to Equation (5), does the algorithm require scanning all data?
5. Is the probability in Equation (2) known or assumed?

---

> ### Author Response · Authors · 2024-11-15
> **Rebuttal by Authors**
>
> Thank you for your constructive review. We have revised the paper to address your comments faithfully, and highlighted the revised parts in blue. Please find our responses below.
> ***
>
> > **Comparison to classification methods**
>
> Unlike classification tasks, data in rank estimation have "ordered" classes. In other words, labels of ordered data are continuous, thus the difference between classes can be very small. As stated in L41-42, classification methods suffer from discriminating subtle differences across ordered classes. The table below compares the proposed algorithm to recent classification methods on the MORPH II dataset. The proposed SOL significantly outperforms the classification methods in noisy rank estimation tasks. We have included these comparisons in the revised paper. Please see Table 1 on page 6 and Tables 12, 13, 14 on page 16 and 17.
>
> |          | $\kappa=0.2$ |              | $\kappa=0.3$      |                  | $\kappa=0.4$      |                  | $\kappa=0.5$      |                  |
> |--------------------|:-------------------:|:------------------:|:-------------------:|:------------------:|:-------------------:|:------------------:|:-------------------:|:------------------:|
> | Algorithm          | MAE | CS| MAE| CS | MAE| CS | MAE| CS |
> | SPR [1]               | 8.446             | 41.71            | 8.881             | 34.79            | 9.239             | 36.89            | 9.993             | 28.14            |
> | ACL [2]               | 9.017             | 36.75            | 9.492             | 35.61            | 9.314             | 35.74            | 9.743             | 34.16            |
> | SOL w/o refinement | 2.507  | 91.26| **2.657**         | 88.71            | 2.842 | 86.89            | 2.995 |84.79|
> | SOL                | **2.489**         | **91.35**        |2.663 | **89.62**        | **2.826**         | **87.70**        | **2.986**         | **85.88**        |
>
>
> [1] Scalable penalized regression for noise detection in learning with noisy labels. In CVPR, 2022.
>
> [2] Active negative loss functions for learning with noisy labels. In NIPS, 2023.
>
> > **Writing**
>
> With all due respect, we do not agree with the reviewer. Please provide specific examples. We will do our best to revise the paper.
>
> > **Definition of $d(\cdot,\cdot)$**
>
> In L156, $d(\cdot, \cdot)$ is defined. It denotes the Euclidean distance between two points in the embedding space.
>
> > **Noise range**
>
> In real-world problems, generally, the range of true labels is unknown. Hence, we do not impose constraints to ensure that $r_x$, the rank given after noise $e_x$ is added, is within the range of true labels in Eq.(1). However, we clip the values so that $r_x \geq 0$, to avoid impractical (i.e.  negative) ranks.
>
> > **Relationship of $\mu_{r_x+s}$ and $p_s$**
>
> If the label noise of instance $x$ is $s$, the true rank of $x$ would be equal to $r_x+s$. Since $p_s$ denotes the probability that the label error of $x$ is $s$, $x$ is associated with $\mu_{r_x+s}$ with probability $p_s$. If we have misunderstood your comment on clarifying the relationship between $\mu_{r_x+s}$ and $p_s$, please let us know and we will address them more faithfully.
>
> > **Equation (5)**
>
> Yes, all data is scanned when computing the centroids in Eq.(5).
>
> > **Probability in Equation (2)**
>
> As stated in L137, the random variable $\mathbf{e}$ is assumed to have a discrete Gaussian distribution. Thus, the probability in Eq.(2) is assumed.
>
> ***
> If you have any additional concerns, please let us know. We will address them faithfully. Thank you again for your feedback.

---

> > ### Comment · Reviewer_nnr1 · 2024-11-25
> >
> > Thank you for the authors' responses. My concerns are as follows:
> >
> > 1. While the authors demonstrate subtle differences between ordered learning and classification problems, I still believe that their training processes are fundamentally similar. Could the authors clarify why classification algorithms cannot address ordered learning, or why ordered learning is necessary? Additionally, this aspect seems to be missing in the current submission.
> >
> > 2. The proposed method appears to have limited effectiveness. Since scanning the entire dataset is required in the proposed algorithm, it is inefficient. Especially, when dealing with large volumes of data.
> >
> > 3. In Equation 2, could the authors clarify how the variance $\sigma$ of the Gaussian distribution is obtained?

---

> ### Author Response · Authors · 2024-11-26
>
> Thank you for your feedback. We appreciate it greatly.
> ***
>
> > **Difference from classification**
>
> Rank estimation (or ordinal regression), which order learning algorithms aim to solve, is fundamentally different from ordinary classification. Unlike classification, different errors have different severities in rank estimation; misclassifying a 20-year-old as a 10-year-old is severer than mistaking it for a 22-year-old. Thus, for reliable rank estimation, it is important to avoid such severe errors. However, classification-based approaches do not consider this ordinal property in rank estimation datasets, and thus yield poor performances when applied to noisy rank estimation tasks. We clarified this by adding more explanation in the introduction of the revised paper.
>
> Also, the training method of SOL is different from that of classification methods such as SPR and ACL. While they employ cross-entropy based losses which equally treat all noise (or label deviation), SOL exploits the discriminative loss and the stochastic order loss to leverage the ordinal relationship of instances/classes. Experimental results including Table 1, 12, 13, and 14 show that SOL outperforms classification-based approaches by meaningful margins.
>
> > **Efficiency**
>
> Below, we measure the time required to compute the centroids in Eq.(5). We use an RTX 4090 GPU. Even for the RSNA dataset which consists of 12,611 training samples, it takes only a few minutes. It is fast enough for most use cases, since centroid update is performed only once at each epoch. Thus, as shown in the table below, the centroid update accounts for only a small portion of the overall training time. Moreover, even with very large datasets, we can easily reduce the computation time by subsampling the training instances for the centroid update.
>
>
> |                                     |      MORPH II  |  CLAP2015   |      AADB      |  RSNA   |
> |:----------------------------- |:-------------------:|:-----------------:|:-----------------:|:-----------------:|
> | Centroid computation |      6.10s          |       5.08s       |      39.18s     |      286.06s    |
> |       Training 1 epoch   |      60.17s        |      52.07s       |     145.43s | 1160.66s      |
>
>
> For large scale training, memory efficiency is also important. We compare the number of parameters of SOL with that of conventional methods. SOL requires the smallest number of parameters, showing its potential for large scale applications. We have included these results in the revised paper. Please see Tables 17 and 18 in the Appendix.
>
> |                      |     ACL      |      MWR    |  GOL  |SOL   |
> |:-------------------- |:------------------:|:-----------------:|:-----------------:|:-----------------:|
> | # of parameters |     134.68M        |       139.41M       |    14.75M       |      14.72M       |
>
>
> > **$\sigma$ in Eq.(2)**
>
> In practice, prior knowledge of noise is not given. To simulate such practical settings, we also assume that noise information is unknown. Hence, $\sigma$ in Eq. 2 is a hyper-parameter in SOL and determined empirically. We have already provided the analysis of $\sigma$ in Table 9 (also in the table below). In general, SOL provides decent results at $\sigma = 1$ regardless of ground-truth noise variance. Therefore, we use $\sigma=1$ as our default option for all experiments.
>
> |                | $\sigma=0.5$ || $\sigma=1$ || $\sigma=1.5$||
> |:---------------|:----------------:|:----------------:|:------------------:|:--------------:|:---------------:|:-------------------:|
> |                | MAE               | CS              | MAE              | CS  | MAE | CS  |
> | $\kappa=0.2$ | 2.557                            | 90.62                          | 2.507                            | 91.26           | 2.529              | 90.80           |
> | $\kappa=0.3$ | 2.682                            | 88.62                          | 2.657                            | 88.71           | 2.689              | 89.44           |
> | $\kappa=0.4$ | 2.850                            | 86.98                          | 2.842                            | 86.89           | 2.869              | 86.70           |
>
>
> ***
> If you have any additional concerns, please let us know. We will do our best to resolve them.

---

### Official Review · Reviewer_2xn2 · 2024-11-03

**Soundness:** 2
**Presentation:** 3
**Contribution:** 3
**Rating:** 6
**Confidence:** 3

**Summary:**

To solve the rank estimation problem with label noise, this manuscript proposes the stochastic order learning (SOL) method. Specifically, label errors are represented as random variables that follow a Gaussian distribution. Based on this, the stochastic dissimilarity of instance x from rank r is defined and the corresponding discriminative loss and stochastic order loss are designed to train a network to build an embedding space where instances are arranged according to their ranks. Furthermore, after network training, the SOL method identifies outliers that are likely to have extreme label errors and relabel them to refine the dataset.

**Strengths:**

1. This manuscript models label errors as stochastic variables and derives embedding space constraints to arrange instances based on their probabilistically associated ranks.
2. This manuscript proposes outlier detection and relabeling schemes to pinpoint and mitigate instances with extreme label errors, thereby diminishing the dataset's overall noise level.

**Weaknesses:**

1. The reason for setting formulas 12-14 lacks description. Specifically, why is the formula defined in this way? It is necessary to explain the meaning of setting $r_x-r_y-s+t$. In other words, whether I can consider that if the label noise of samples x and y is s and t, respectively, the distance between the true labels of the two samples is $r_x-r_y-s+t$.
2. In the experiments, there are few comparative methods and no comparison with the state-of-the-art methods. If there are few recent papers about rank estimation with label noise, it can be compared with related methods mentioned in this paper. For example, if rank estimation is regarded as a classification task, is the SOL method better than the approach in the paper [1]? Besides, on datasets with a rank less than 5, whether the SOL method is still performing significantly and better than the method in the paper [2].

[1] Xichen Ye, Xiaoqiang Li, Songmin Dai, Tong Liu, Yan Sun, and Weiqin Tong. Active negative loss functions for learning with noisy labels. In NIPS, 2023.

[2] Huan Liu, Jiankai Tu, Anqi Gao, and Chunguang Li. Distributed robust support vector ordinal regression under label noise. Neurocomputing, pp. 128057, 2024.

**Questions:**

1. The proposed method settings tend to have a risk of labeling errors for each sample. More likely, only a small amount of samples in the dataset is mislabeled. In this situation, is the performance of the method still effective? In addition, the proposed method is based on the key assumption that the label noise of each sample follows the same discrete Gaussian distribution. It is more natural that the standard deviation of noise variables varies among different samples. In the future, a fine-tuning mechanism for setting standard deviations for each sample can be considered while taking into account computational complexity.
2. There are many hyperparameters in the manuscript. Can an adaptive mechanism be designed to select the optimal values of these parameters for different datasets?
3. In the relabeling mechanism, the rank of each detected outlier is adjusted by the same magnitude, that is, half of the average absolute difference between noisy and estimated rankings overall training instances, which seems counterintuitive. It is reasonable to combine outlier detection with the relabeling mechanism to adjust the rank of the outlier.

---

> ### Author Response · Authors · 2024-11-15
> **Rebuttal by Authors**
>
> Thank you for your positive review and insightful suggestions. We have revised the paper to address your comments faithfully, and highlighted the revised parts in blue. Please find our responses below.
> ***
>
> > **Formulas (12)~(14)**
>
> As you pointed out, the difference between the true labels of $x$ and $y$ is $r_x-r_y-s+t$ when the label noise of $x$ and $y$ is $s$ and $t$, respectively. Following Eq.(11), the ordering relationship is $x \prec y$ when $r_x-r_y-s+t < -\tau$. Since we model label noise as stochastic variables, we can compute the probability of $x \prec y$ as Formula (12) by using Eq.(2). We have clarified this point in L235-239 of the revised paper.
>
> > **More comparisons**
>
> As suggested, we compare our algorithms with more noise-robust methods [1,3,4] on the MORPH II, CLAP2015, AADB, and RSNA datasets. The table below shows the comparison results on the MORPH II dataset. The proposed SOL still achieves the best scores. The results for other three datasets are similar as well. We have included all these results in the revised paper. Please see Table 1 on page 6 and Tables 12, 13, 14 on page 16 and 17.
>
> For method RSVOR[2], we were not able to re-implement it successfully — RSVOR yields very poor results only — because its source code is not available and the authors haven’t responded to our question about implementation details. Moreover, RSVOR is designed for tabular datasets which are very different from vision datasets used in SOL. Hence, the application of RSVOR is not straightforward. We will try to fine-tune the implementation of RSVOR as much as possible and update the results.
>
> ||$\kappa=0.2$||$\kappa=0.3$||$\kappa=0.4$||$\kappa=0.5$||
> |-|:-:|:-:|:-:|:-:|:-:|:-:|:-:|:-:|
> | Algorithm|MAE|CS|MAE|CS|MAE|CS| MAE|CS|
> |ACL [1]|9.017|36.75|9.492|35.61|9.314|35.74|9.743|34.16|
> |SPR [3]|8.446|41.71|8.881|34.79|9.239|36.89|9.993|28.14|
> |C-Mixup [4]|3.063|82.26|3.393|77.21|3.395|76.84|3.415|76.65|
> |SOL w/o refinement|2.507|91.26|**2.657**|88.71|2.842|86.89|2.995|84.79|
> |SOL|**2.489**|**91.35**|2.663|**89.62**|**2.826**|**87.70**|**2.986**|**85.88**|
>
> [1] Active negative loss functions for learning with noisy labels. In NIPS, 2023.
>
> [2] Distributed robust support vector ordinal regression under label noise. Neurocomputing, pp. 128057, 2024.
>
> [3] Scalable penalized regression for noise detection in learning with noisy labels. In CVPR, 2022.
>
> [4] C-Mixup: Improving generalization in regression. In NIPS, 2022.
>
> > **Partially corrupted data**
>
> We measured the performances of SOL and GOL when only a small amount of samples are mislabeled. In this test, we use the CLAP2015 dataset and randomly select 10% of samples for label corruption. As shown in the table below, SOL is still effective even when a small amount of samples have label errors. We have added these results in the revised paper. Please see Table 15 and its description on page 17.
>
> |Algorithm|$\kappa=0.2$|$\kappa=0.3$|$\kappa=0.4$|$\kappa=0.5$|
> |-|-|-|-|-|
> |GOL|3.442|3.540|3.590|3.690|
> |SOL|**3.420**|**3.505**|**3.549**|**3.639**|
>
> > **Noise distribution**
>
> We agree with you. Please note that SOL is the first attempt to extend the concept of order learning to noise-robust rank estimation, which is a relatively under-researched area. Hence, we focus on developing a simple but reasonable algorithm instead of covering all details in real-world scenarios. It is a future research issue to set the noise distribution adaptively according to each instance. We have clarified this point in the revised paper. Please see the last paragraph on page 17.
>
> > **Hyper-parameters**
>
> We observed that as long as the hyper-parameters do not deviate much from the proposed default settings, decent performances are yielded. In Table 10 of page 14, we listed all the hyper-parameter values used for all datasets. Please note that the values do not vary much for the different datasets. We do agree, however, that designing an adaptive mechanism may be helpful for improving the scalability of SOL to various datasets. We will investigate it in future work. Thank you for your constructive comment.
>
> > **Relabeling mechanism**
>
> During the development of SOL, we have tried a relabeling scheme where outliers are relabeled by different magnitudes. The table below compares the performances according to the relabeling scheme on CLAP2015 at $\kappa=0.4$. Although relabeling with different magnitudes also show performance improvement, the proposed technique yields better results. It is because the proposed relabeling scheme prevents drastic rank changes, as stated in L314. Please see Table 16 on page 17 for more details.
>
> |Relabeling mechanism|MAE|CS|
> |-|-|-|
> |No relabeling|4.058|**73.68**|
> |Different magnitudes|4.012|72.75|
> |Proposed|**4.002**|**73.68**|
>
> ***
>
> We have made every attempt to address your comments in the revised paper. We hope that you find this revision satisfactory. If you have additional concerns, please let us know. Thank you again for your positive comments.

---

> > ### Comment · Reviewer_2xn2 · 2024-11-17
> >
> > Thank you for the response. Most of my questions have been clarified, and I will keep my score.

---

> > > ### Author Response · Authors · 2024-11-17
> > >
> > > Thank you for your time and effort for reviewing our paper. We appreciate your positive comments and review.

---

### Author Response · Authors · 2024-11-13

We would like to thank all reviewers for their time and efforts for providing constructive reviews. We will upload our response to each question or comment as soon as possible.

---

### Meta-Review · Area_Chair_ccAz · 2024-12-19

**Metareview:**

Thanks for your submission to ICLR.

This paper received four reviews, which were borderline tending to reject.  On the positive side, the reviewers appreciated that this paper addresses a practical problem.  But they had various concerns, including i) issues with the presentation / missing descriptions, ii) issues with the experiments, including missing comparisons to other methods, iii) concerns about the effectiveness / novelty of the approach.

The author response clarified some of these issues, but the negative reviewers were unwilling to raise their scores.  I think the rebuttal did address several of these issues, but I also think (particularly given the overall lack of enthusiasm with the current paper) that the paper could use some additional refinement to address these issues.  With more work it seems that the paper could definitely be publishable.

**Additional Comments On Reviewer Discussion:**

Two of the reviewers participated in the discussion, but two did not.  The more negative reviewer who participated maintained their score after the discussion period.  I took a look at the concerns of the other reviewers, as well as the rebuttal.  Though the rebuttal did resolve some issues, I think the paper still needs work before it is ready to be published.

---

### Decision · Program_Chairs · 2025-01-22

Reject